# Efficacy and safety of vamorolone in Duchenne muscular dystrophy: An 18-month interim analysis of a non-randomized open-label extension study

Edward C. Smith[1], Laurie S. Conklin[2,3], Eric P. Hoffman[2,4], Paula R. Clemens[5], Jean K. Mah[6], Richard S. Finkel[7], Michela Guglieri[8], Mar Tulinius[9], Yoram Nevo[10], Monique M. Ryan[11], Richard Webster[12], Diana Castro[13], Nancy L. Kuntz[14], Laurie Kerchner[15], Lauren P. Morgenroth[15], Adrienne Arrieta[15], Maya Shimony[15], Mark Jaros[16], Phil Shale[16], Heather Gordish-Dressman[3], Laura Hagerty[2], Utkarsh J. Dang[4], Jesse M. Damsker[2], Benjamin D. Schwartz[17], Laurel J. Mengle-Gaw[17], Craig M. McDonald[18]*, the CINRG VBP15 and DNHS Investigators

1 Duke University, Durham, North Carolina, United States of America, 2 ReveraGen Biopharma, Rockville, Maryland, United States of America, 3 Children's National Hospital, Washington, District of Columbia, United States of America, 4 Binghamton University–SUNY, Binghamton, New York, United States of America, 5 University of Pittsburgh and Department of Veterans Affairs Medical Center, Pittsburgh, Pennsylvania, United States of America, 6 Alberta Children's Hospital, Calgary, Alberta, Canada, 7 Nemours Children's Hospital, Orlando, Florida, United States of America, 8 John Walton Muscular Dystrophy Research Centre, Newcastle University, Newcastle upon Tyne Hospitals NHS Foundation Trust, Newcastle upon Tyne, United Kingdom, 9 Queen Silvia Children's Hospital, Gothenburg, Sweden, 10 Schneider Children's Medical Center, Tel Aviv University, Petah Tikvah, Israel, 11 Royal Children's Hospital and Murdoch Children's Research Institute, Melbourne, Victoria, Australia, 12 The Children's Hospital at Westmead, Sydney, New South Wales, Australia, 13 University of Texas Southwestern Medical Center, Dallas, Texas, United States of America, 14 Ann & Robert H. Lurie Children's Hospital of Chicago, Chicago, Illinois, United States of America, 15 TRiNDS, Pittsburgh, Pennsylvania, United States of America, 16 Summit Analytical, Denver, Colorado, United States of America, 17 Camden Group, St. Louis, Missouri, United States of America, 18 University of California, Davis, Davis, California, United States of America

* cmmcdonald@ucdavis.edu

**Data Availability Statement:** All relevant data are within the manuscript and its Supporting Information files.

## Abstract

### Background

Treatment with corticosteroids is recommended for Duchenne muscular dystrophy (DMD) patients to slow the progression of weakness. However, chronic corticosteroid treatment causes significant morbidities. Vamorolone is a first-in-class anti-inflammatory investigational drug that has shown evidence of efficacy in DMD after 24 weeks of treatment at 2.0 or 6.0 mg/kg/day. Here, open-label efficacy and safety experience of vamorolone was evaluated over a period of 18 months in trial participants with DMD.

### Methods and findings

A multicenter, open-label, 24-week trial (VBP15-003) with a 24-month long-term extension (VBP15-LTE) was conducted by the Cooperative International Neuromuscular Research Group (CINRG) and evaluated drug-related effects of vamorolone on motor outcomes and corticosteroid-associated safety concerns. The study was carried out in Canada, US, UK, Australia, Sweden, and Israel, from 2016 to 2019. This report covers the initial 24-week trial and the

**Funding:** This work was funded by: National Institutes of Health NINDS R44NS095423 [EPH, PRC] www.ninds.nih.gov. National Institutes of Health NICHD 5U54HD090254 [EPH, LSC] www.nichd.nih.gov. National Institutes of Health NIAMS U34AR068616 [PRC], www.niams.nih.gov. European Commission Horizons 2020 grant agreement number 667078 [MG], https://cordis.europa.eu/project/id/667078. The funders had no role in study design, data collection and analysis, decision to publish, or preparation of the manuscript.

**Competing interests:** I have read the journal's policy and the authors of this manuscript have the following competing interests: LSC, JMD, LH and EPH are employees of ReveraGen BioPharma. LSC, JMD, LH own stock options of ReveraGen. UD is a paid consultant for ReveraGen. MJ and PS are employees of Summit Analytical, a biostatistics clinical research organization. BDS and LJM G own Camden Group, LLC, a clinical research organization. EPH and HG-D are co-founders and members of the Board, and ALD'A, LPM, AA, and MS are employees of TRiNDS LLC, a clinical trials management organization. PRC, ECS, JKM, RSF, MG, MT, YN, MMR, RW, DC, NLK, and CMM have received grant funding from ReveraGen for the conduct of clinical trials but they have not received compensation from ReveraGen for other activities. CMM has served as a consultant for clinical trials in Duchenne muscular dystrophy outside the submitted work for Astellas, Biomarin, Capricor Therapeutics, Cardero Therapeutics, Inc., Catabasis Pharmaceuticals, Eli Lilly, FibroGen, Marathon Pharmaceuticals, Pfizer, PTC Therapeutics, Santhera Pharmaceuticals, Sarepta Therapeutics, PTC Therapeutics; serves on external advisory boards related to Duchenne muscular dystrophy for PTC Therapeutics, Eli Lilly, Sarepta Therapeutics, Santhera Pharmaceuticals, and Capricor; and reports grants US Dept. of Education/NIDRR, NIDILRR, US NIH/NIAMS, US Dept. of Defense, and Parent Project Muscular Dystrophy US, during the conduct of the study. Patents awarded relevant to the results include: WO2017004205 (A1), US2016060289 (A1), US2015011519 (A1), US9649320 (B2); US2017027959 (A1). The authors have declared that no other competing interests exist.

**Abbreviations:** 6MWT, 6-minute walk test; AE, adverse event; CINRG, Cooperative International Neuromuscular Research Group; DMD, Duchenne muscular dystrophy; DNHS, Duchenne Natural History Study; iSAP, interim statistical analysis plan; LS, least squares; MAD, multiple ascending dose; NSAA, North Star Ambulatory Assessment;

first 12 months of the VBP15-LTE trial (total treatment period 18 months). DMD trial participants (males, 4 to <7 years at entry) treated with 2.0 or 6.0 mg/kg/day vamorolone for the full 18-month period ($n = 23$) showed clinical improvement of all motor outcomes from baseline to month 18 (time to stand velocity, $p = 0.012$ [95% CI 0.010, 0.068 event/second]; run/walk 10 meters velocity, $p < 0.001$ [95% CI 0.220, 0.491 meters/second]; climb 4 stairs velocity, $p = 0.001$ [95% CI 0.034, 0.105 event/second]; 6-minute walk test, $p = 0.001$ [95% CI 31.14, 93.38 meters]; North Star Ambulatory Assessment, $p < 0.001$ [95% CI 2.702, 6.662 points]). Outcomes in vamorolone-treated DMD patients ($n = 46$) were compared to group-matched participants in the CINRG Duchenne Natural History Study (corticosteroid-naïve, $n = 19$; corticosteroid-treated, $n = 68$) over a similar 18-month period. Time to stand was not significantly different between vamorolone-treated and corticosteroid-naïve participants ($p = 0.088$; least squares [LS] mean 0.042 [95% CI –0.007, 0.091]), but vamorolone-treated participants showed significant improvement compared to group-matched corticosteroid-naïve participants for run/walk 10 meters velocity ($p = 0.003$; LS mean 0.286 [95% CI 0.104, 0.469]) and climb 4 stairs velocity ($p = 0.027$; LS mean 0.059 [95% CI 0.007, 0.111]). The vamorolone-related improvements were similar in magnitude to corticosteroid-related improvements. Corticosteroid-treated participants showed stunting of growth, whereas vamorolone-treated trial participants did not ($p < 0.001$; LS mean 15.86 [95% CI 8.51, 23.22]). Physician-reported incidences of adverse events (AEs) for Cushingoid appearance, hirsutism, weight gain, and behavior change were less for vamorolone than published incidences for prednisone and deflazacort. Key limitations to the study were the open-label design, and use of external comparators.

## Conclusions

We observed that vamorolone treatment was associated with improvements in some motor outcomes as compared with corticosteroid-naïve individuals over an 18-month treatment period. We found that fewer physician-reported AEs occurred with vamorolone than have been reported for treatment with prednisone and deflazacort, and that vamorolone treatment did not cause the stunting of growth seen with these corticosteroids. This Phase IIa study provides Class III evidence to support benefit of motor function in young boys with DMD treated with vamorolone 2.0 to 6.0 mg/kg/day, with a favorable safety profile. A Phase III RCT is underway to further investigate safety and efficacy.

## Trial registration

Clinical trials were registered at www.clinicaltrials.gov, and the links to each trial are as follows (as provided in manuscript text): VBP15-002 [NCT02760264] VBP15-003 [NCT02760277] VBP15-LTE [NCT03038399].

## Author summary

### Why was this study done?

- The standard-of-care pharmacological management of Duchenne muscular dystrophy (DMD) is high-dose corticosteroids (prednisone or deflazacort; about 0.5 to 0.9 mg/kg/day), but this treatment is associated with safety concerns.

PK, pharmacokinetics; TEAE, treatment-emergent adverse event; TTCLIMB, time to climb 4 stairs; TTRW, time to run/walk 10 meters; TTSTAND, time to stand from supine.

- Vamorolone is a first-in-class steroidal drug that aims to retain or improve efficacy of corticosteroids, while decreasing safety concerns.
- Twenty-four-week treatment of DMD with vamorolone has been reported to show dose-responsive improvements in motor outcomes, but longer-term treatment and comparative safety profiles to corticosteroids have not been previously reported.

## What did the researchers do and find?

- Participants with DMD completing the 24-week dose-ranging study of vamorolone (VBP15-003; $n = 46$) were offered transition to standard of care (prednisone or deflazacort) or continued treatment with vamorolone with enrollment into a 2-year long-term extension study (VBP15-LTE). All participants (46/46) opted to continue treatment with vamorolone and enrolled in VBP15-LTE. We report data from the midpoint of the 2-year VBP15-LTE study (total of 18 months of vamorolone treatment in VBP15-003 + VBP15-LTE). All measures of efficacy (5 motor outcome tests) showed significant improvements from baseline to 18 months of vamorolone treatment by paired intragroup analyses.
- Three motor outcome tests could be compared to group-matched corticosteroid-naïve external comparators, and 2 showed significant vamorolone-associated improvement.
- Vamorolone treatment did not result in stunting of growth, as seen with prednisone and deflazacort. Vamorolone treatment showed fewer physician-reported adverse events (such as behavior change, hirsutism, and Cushingoid appearance) than these drugs.

## What do these findings mean?

- Vamorolone holds promise as a replacement for corticosteroid standard of care (prednisone and deflazacort) in DMD.
- Limitations of the study include the open-label design and use of external comparators.

## Introduction

Vamorolone is an anti-inflammatory steroidal drug that differs from all 33 drugs in the corticosteroid class by lacking an 11-carbon oxygen group (hydroxyl or carbonyl) that is 1 of 5 molecular contact sites with the glucocorticoid receptor [1,2]. In vitro pharmacology and pre-clinical in vivo studies have shown that vamorolone retains the anti-inflammatory activity of steroid drugs, while lacking the adverse effects (AEs) associated with these drugs (stunting of growth, bone morbidities, muscle atrophy) in these models [3,4]. Many corticosteroids, including prednisone and deflazacort, are agonists of the mineralocorticoid receptor, leading to increased blood volume and pressure via the renin–angiotensin pathway. In contrast, vamorolone is a potent antagonist of the mineralocorticoid receptor, similar in activity to eplerenone and spironolactone [5]. The differential mechanism of action of vamorolone compared to traditional corticosteroid anti-inflammatory drugs is attributed to the loss of gene

transcriptional activities associated with glucocorticoid response element binding and activation, potent antagonist activity for the mineralocorticoid receptor, superior membrane stabilization properties, and retention of the distinct NFκB inhibitory (anti-inflammatory) activities [3,5–7]). Activation of NFκB-related cell damage pathways is recognized as one of the earliest molecular pathologies of dystrophin-deficient muscle in Duchenne muscular dystrophy (DMD) patients, and both vamorolone and corticosteroids inhibit these pathways [8].

Vamorolone clinical studies have been conducted in adult male volunteers [9], and in boys with DMD, a disorder in which skeletal muscle is in a chronic inflammatory state [10]. Two consecutive open-label dose-ranging studies in 48 DMD patients aged 4 to <7 years (corticosteroid-naïve) were conducted (Phase IIa, VBP15-002; Phase IIa, VBP15-003). Doses were tested over a 24-fold dose range (0.25, 0.75, 2.0, and 6.0 mg/kg/day), with 12 participants per group. The first multiple ascending dose (MAD) cohort trial tested pharmacokinetics (PK) and safety for 2 weeks of drug dosing followed by a 2-week washout (VBP15-002) [11]. Vamorolone treatment in this study showed no dose-limiting toxicities, and PK demonstrated a short half-life similar to corticosteroids (~2 hours), no drug accumulation, similar PK on day 1 and day 14, and PK similar to that of healthy adult male volunteers (VBP15-001) [9,11–13]. All DMD participants completed the MAD study and then continued on the same dose for a 24-week dose-finding (efficacy and safety) extension study (VBP15-003). Oral administration of vamorolone at all doses tested was safe and well tolerated over the 24-week treatment period. Participants in the 2 higher dose groups (2.0 and 6.0 mg/kg/day) generally showed clinical improvement of motor outcomes, with suggestion of dose-related improvements in all motor outcomes tested [2,14]. However, the 24-week trial period did not allow for adequate evaluation of adverse effects associated with longer-term chronic corticosteroid exposure in children. A 24-week study also does not allow for assessing the longer-term efficacy of vamorolone.

After completion of the 24-week dose-finding study (VBP15-003), participants had the opportunity to enroll in a 24-month long-term extension study (VBP15-LTE) that permitted dose escalations and de-escalations. All trial participants' parents and physicians requested continued access to vamorolone, rather than transition to standard of care (prednisone or deflazacort). Here, we report the initial experience from the 24-week VBP15-003 trial and the first 12 months of the 24-month VBP15-LTE trial (total 18 months of treatment). Change in motor function and safety outcomes are also compared to data from group-matched corticosteroid-treated and corticosteroid-naïve participants enrolled in the Cooperative International Neuromuscular Research Group (CINRG) Duchenne Natural History Study (DNHS) [15,16]. Safety endpoints (linear growth, body mass index) are also compared with data from a 12-month trial of daily prednisone (0.75 mg/kg group) in similar-aged boys with DMD [17].

## Methods

### Ethics statement

All studies had appropriate approvals by ethics committees or institutional review boards, as required by the 11 participating international academic clinical recruitment sites: (Duke University, Durham, NC, US; Alberta Children's Hospital, Calgary, AB, Canada; Nemours Children's Hospital, Orlando, FL, US; John Walton Muscular Dystrophy Research Centre, Newcastle University, Newcastle upon Tyne Hospitals NHS Foundation Trust, Newcastle upon Tyne, UK; Queen Silvia Children's Hospital, Gothenburg, Sweden; Schneider Children's Medical Center, Tel Aviv University, Petah Tikvah, Israel; Royal Children's Hospital and Murdoch Children's Research Institute, Melbourne, VIC, Australia; The Children's Hospital at Westmead, Sydney, NSW, Australia; University of Texas Southwestern Medical Center, Dallas, TX, US; Ann & Robert H. Lurie Children's Hospital of Chicago, Chicago, IL, US; University of California Davis, Davis,

CA, US). Informed written consent was obtained from the parents (or guardians) of each child recruited into the described trials. Separate ethics review and written informed consent were obtained for the 3 consecutive trials (VBP15-002, VBP15-003, VPB15-LTE).

## Trial participants

Three consecutive clinical trials of vamorolone treatment of DMD were conducted by CINRG (VBP15-002 [NCT02760264]; VBP15-003 [NCT02760277]; VBP15-LTE [NCT03038399]). A total of 48 participants (ages 4 to <7 years) were initially enrolled into VBP15-002, with a study start date of June 2016, with trial participants completing month 12 of the 24-month VBP15-LTE study by April 2019. Participants were recruited in the US, Canada, UK, Sweden, Israel, and Australia.

VBP15-002 (Phase IIa; 2 weeks on drug, 2 weeks off drug) enrolled 48 corticosteroid-naïve participants with DMD, and all 48 participants completed the study and enrolled into VBP15-003 (Phase IIa extension; 24-week treatment). Forty-six of 48 participants completed the VBP15-003 study (2 participants withdrew from VBP15-003 for reasons not related to study drug) [14]. All participants (46/46) opted to enroll in the 24-month long-term extension study, VBP15-LTE. This current report focuses on the 12-month interim time point in the 24-month VBP15-LTE protocol (S1 Protocol). This study is reported as per the Transparent Reporting of Evaluations with Nonrandomized Designs (TREND) guideline (S1 TREND Checklist).

The consecutive vamorolone trials (VBP15-002, VBP15-003, VBP15-LTE) were open-label with no placebo comparator. Corticosteroid-naïve and corticosteroid-treated DMD participant comparators were group-matched participants from the CINRG DNHS (NCT00468832). The CINRG DNHS was an observational, prospective case–control study of 551 participants (440 with DMD, 111 healthy peers), with a study start date of December 2005, and study completion date of November 2016 [15,16]. For group matching between vamorolone-treated participants and CINRG DNHS participants, pre-specified criteria were defined for matching within the interim statistical analysis plan (iSAP) (S1 iSAP). Two statisticians independent of the sponsor applied the matching criteria to the CINRG DNHS cohort based on age and other characteristics (S1 Table), and the selected participants were harmonized and agreed upon with blinding to outcome data. Age-matched CINRG DNHS participants included those continuously corticosteroid-naïve over an 18-month period ($n$ = 19) or continuously corticosteroid-treated over an 18-month period ($n$ = 68). For the 68 corticosteroid-treated participants, as this was an observational cohort, corticosteroid doses and regimens varied based on clinician discretion [18]. Although all 68 participants were treated for 18 months continuously, the age at initiation of corticosteroids varied. Thus, the total duration of corticosteroid treatment was longer than 18 months for most participants.

For comparisons of growth trajectories of vamorolone- and corticosteroid-treated participants, we utilized a third external comparator of a CINRG 12-month prednisone clinical trial (daily treated arm, 0.75 mg/kg/day) [17]. As with the CINRG DNHS comparators, group-matching criteria were pre-specified in the iSAP, and 2 independent statisticians carried out the participant matching. The efficacy data from the CINRG 12-month prednisone trial were not compared to those of the vamorolone-treated participants, as there was no corresponding 12-month assessment in vamorolone-treated participants (assessments of vamorolone-treated trial participants were at 0, 3, 6, and 18 months).

## Measurements

Assessments of efficacy were motor outcomes (primary outcome: time to stand from supine [TTSTAND]; secondary outcomes: time to run/walk 10 meters [TTRW], time to climb 4 stairs

[TTCLIMB], distance covered in 6-minute walk test [6MWT], and the North Star Ambulatory Assessment [NSAA]). 6MWT and NSAA were not assessed in the majority of CINRG DNHS participants and were not compared to vamorolone-treated participants. Clinical evaluators were trained according to standard operating procedures that were harmonized between the CINRG vamorolone, CINRG DNHS, and CINRG prednisone studies. Reliability of these outcomes (percent coefficient of variation) has been reported for the VBP15-002/VBP15-003 studies [19]. Assessments were done at baseline (VBP15-002 entry), 24 weeks (VBP15-003 last visit), and 18 months (VBP15-LTE midpoint assessment at 12 months).

Standing height and weight were assessed at each study visit. Height z-score, body mass index (BMI; kg/m$^2$), and BMI z-score were calculated centrally. AE reporting was done per protocol in the vamorolone trials.

## Study design

Only participants completing VBP15-002 and VBP15-003 were eligible to enroll in VBP15-LTE. Participants received vamorolone at 1 of 4 dose levels (0.25, 0.75, 2.0, or 6.0 mg/kg/day), and at the same dose level in both the 4-week VBP15-002 trial and the 24-week VBP15-003 trial. If participants, their families, and their physician wished to continue vamorolone treatment upon exiting the VBP15-003 trial, they were offered participation in the 24-month long-term extension (VBP15-LTE). The last visit of the VBP15-003 trial was commensurate with the first visit of the VBP15-LTE trial. In all studies, study medication was provided as 4% flavored liquid suspension and was dosed according to body weight and given once daily in the morning with food.

Study visits took place quarterly, including assessment of clinical laboratory results, vital signs, and AEs. All AEs were coded using the Medical Dictionary for Regulatory Activities (MedDRA version 19.0) system for reporting (preferred term and system organ class). Clinical efficacy assessments were performed at baseline of the VBP15-002 study, at 6 months (end of VBP15-003 study), and at the 12-month midpoint visit of the VBP15-LTE study.

The VBP15-LTE protocol permitted multiple dose escalations to the highest dose (6.0 mg/kg/day) at the discretion of the participant's family and physician, and also permitted de-escalations. Site investigators were permitted to escalate a participant's dose to a higher dose level during the VBP15-LTE (6.0 mg/kg/day) with Study Chair and Medical Monitor approval once the participant had been on their initial dose in VBP15-LTE for at least 1 month, the next higher dose was determined to be safe in the VBP15-002 Phase IIa Study, and no safety issues with that dose had emerged in the VBP15-003 Phase IIa study.

Vamorolone-treated participants were initially enrolled into VBP15-002 and VBP15-003 in 4 dose groups (0.25, 0.75, 2.0, and 6.0 mg/kg/day; groups A–D). Upon entering VBP15-LTE, vamorolone group A participants had 2 or 3 sequential dose escalations, and were treated with 2.0 or 6.0 mg/kg/day for the last 3–9 months of the 18-month period; group B participants had 1 or 2 dose escalations and were treated with 2.0 or 6.0 mg/kg/day for the last 9–11 months; and groups C and D were treated for 18 months at 2.0 or 6.0 mg/kg/day (S1 Fig). We kept this group stratification for further analyses not pre-specified in the iSAP, and for cross-sectional comparisons to the same matched corticosteroid-naïve and corticosteroid-treated participants from the CINRG DNHS.

The current study is the first evaluating longer-term tolerability, efficacy, and safety of vamorolone in DMD. The VBP15-003 dose-finding study suggested that vamorolone doses of 2.0 and 6.0 mg/kg/day showed better efficacy and similar safety profiles in comparison to lower doses [14]. Given the variable timing of dose escalations, we pre-specified that initial analyses of drug-related efficacy and safety would be limited to those participants who had 18

months of treatment with 2.0 mg/kg/day vamorolone or more (dose group C + dose group D; $n$ = 23). Outcomes for these participants were compared to a group-matched cohort from the CINRG DNHS over an 18-month period (corticosteroid-naïve, $n$ = 19; corticosteroid-treated, $n$ = 68). Participants were matched for age and treatment period (±1 month), matching criteria were pre-specified in the statistical analysis plan, and matching was carried out by 2 independent statisticians (S1 Table).

Growth trajectories and BMI before/after drug treatment were compared between these CINRG DNHS groups over an 18-month period and were also compared to the cohort of CINRG prednisone clinical trial participants who were treated with daily prednisone for the 12-month treatment period of the trial ($n$ = 12). Participants in the corticosteroid-treated CINRG DNHS group were treated for at least 18 months, but total duration, dose, and regimens varied.

## Statistical analysis

An interim statistical analysis plan was written (VBP15-LTE iSAP) (S1 iSAP). The VBP15-LTE iSAP pre-specified analyses of the VBP15-LTE midpoint (12-month) assessments and comparisons to external comparators (corticosteroid-treated and corticosteroid-naïve participants from CINRG DNHS) [16]. The VBP15-LTE iSAP included all month 12 assessments of the 24-month VBP15-LTE study (ongoing at the time of writing). Software utilized was SAS.

The statistical analyses were carried out in 2 sequential steps. First, we pre-specified groups and comparisons in the VBP15-LTE iSAP. This iSAP included only those vamorolone-treated participants who had been on 2.0 or 6.0 mg/kg/day for the full 18-month treatment period (dose groups C + D), in order to avoid the confounding variable of multiple dose escalations in dose groups A and B (S1 Fig). The second analysis was conducted post hoc after completion of the VBP15-LTE iSAP analyses, with dose stratification based on initial dose group in VBP15-002 (0.25 [group A], 0.75 [group B], 2.0 [group C], and 6.0 [group D] mg/kg/day).

Statistical analyses were done on paired longitudinal outcome data using an ANCOVA approach with change from baseline (VBP15-002) to month 18 (midpoint of VBP15-LTE). Baseline response and age were included as covariates. For the vamorolone-treated participants, age was calculated as (date of informed consent minus birthdate)/365.25. For the DNHS participants, age was calculated as (date of baseline visit used minus birthdate)/365.25. The baseline visit for a DNHS participant was the first visit at which the participant met the comparison eligibility criteria for matching and had a non-missing response for at least 1 endpoint of interest. Timed function tests were analyzed as velocity scores to limit the impact of participants who were unable to perform the test (velocity = 0). Velocity measures are variance-stabilizing transformations, suppressing extreme raw outliers from raw values in seconds; these help with distributional assumptions of the statistical models/tests used. Raw data (seconds) are also reported. Velocity scores for TTSTAND (event/second), TTRW (meters/second), and TTCLIMB (event/second) were imputed as 0 at the first response missing due to inability to perform the test. All other data were observed values only, without imputation. No adjustments for multiplicity on inferential statistics were specified in the iSAP.

For within-group analysis, longitudinal change from baseline to 18 months was analyzed using a paired $t$ test. A longitudinal analysis was not performed for efficacy for the participants in the corticosteroid-treated CINRG DNHS study, as there was no baseline (pre-corticosteroid) efficacy assessment.

## Data access

Raw data used for analyses presented are available in S1 Raw Data.

## Results

### Efficacy of vamorolone on motor function

Demographic and baseline characteristics of the vamorolone-treated and comparator groups are provided in Table 1.

Forty-eight DMD participants were enrolled into VBP15-002 and entered into 4 vamorolone treatment groups (dose group A, 0.25 mg/kg/day; dose group B, 0.75 mg/kg/day; dose group C, 2.0 mg/kg/day; dose group D, 6.0 mg/kg/day). All 48 participants completed the

**Table 1. Demographic and baseline characteristics.**

| Characteristic | VBP15-LTE (group C + D) (*n* = 23) | CINRG DNHS corticosteroid-naïve (*n* = 19) | CINRG DNHS corticosteroid-treated (*n* = 68) | CINRG prednisone trial (*n* = 12) |
|---|---|---|---|---|
| **Age (years)** | | | | |
| Mean | 5.20 | 5.03 | 5.96 | 5.70 |
| SD | 0.90 | 0.55 | 0.64 | 0.66 |
| Median | 4.97 | 4.94 | 6.05 | 5.65 |
| Minimum | 4.01 | 4.02 | 4.25 | 4.80 |
| Maximum | 6.72 | 5.90 | 6.99 | 6.87 |
| **Race, *n* (%)** | | | | |
| Native American | 0 | 0 | 0 | 0 |
| Asian | 0 | 3 (15.8) | 7 (10.3) | 2 (16.7) |
| Black | 0 | 0 | 0 | 0 |
| White | 23 (100) | 15 (78.9) | 56 (82.4) | 8 (66.7) |
| Unknown | 0 | 1 (5.3) | 1 (1.5) | 0 |
| Other | 0 | 0 | 4 (5.9) | 2 (16.7) |
| **Ethnicity, *n* (%)** | | | | |
| Hispanic or Latino | 3 (13.0) | 0 | 5 (7.4) | 1 (8.3) |
| Not Hispanic or Latino | 20 (87.0) | 19 (100) | 63 (92.6) | 11 (91.7) |
| **Weight (kg)** | | | | |
| Mean | 19.5 | 18.3 | 20.6 | 20.1 |
| SD | 2.5 | 2.0 | 3.4 | 3.5 |
| Median | 19.4 | 18.2 | 20.4 | 19.6 |
| Minimum | 15.1 | 15.6 | 15.1 | 16.3 |
| Maximum | 24.0 | 22.3 | 30.3 | 24.8 |
| **Height (cm)** | | | | |
| Mean | 107.0 | 105.4 | 109.2 | 110.3 |
| SD | 6.8 | 5.1 | 5.7 | 6.8 |
| Median | 107.7 | 105.0 | 109.0 | 108.7 |
| Minimum | 95.4 | 97.4 | 96.5 | 102.5 |
| Maximum | 117.5 | 114.0 | 124.3 | 126.5 |
| **Body mass index (kg/m$^2$)** | | | | |
| Mean | 17.0 | 16.4 | 17.2 | 16.5 |
| SD | 0.9 | 0.9 | 1.9 | 1.9 |
| Median | 16.9 | 16.4 | 16.7 | 16.7 |
| Minimum | 15.3 | 14.6 | 14.8 | 13.7 |
| Maximum | 18.6 | 18.3 | 24.2 | 20.0 |

CINRG, Cooperative International Neuromuscular Research Group; DNHS, Duchenne Natural History Study.

4-week VBP15-002 trial, and 46 participants completed the 24-week VBP15-003 trial at the same doses. All 46 participants completing the 24-week VBP15-003 study then opted to enroll into the 24-month long-term extension study (VBP15-LTE) (S1 Fig). Data from the 4-week VBP15-002 study have been published [11], as have data from the 24-week VBP15-003 study [14]. The current study is the first evaluating longer-term tolerability, efficacy, and safety of vamorolone in DMD. The VBP15-003 dose-finding study suggested that the 2 higher vamorolone doses showed greater efficacy than the 2 lower doses.

One participant discontinued the study 1 month before the 12-month assessment (S1 Fig; participant 233504). This participant's 11-month early exit visit data were counted as 12-month study data for this analysis, per the pre-specified iSAP. All participants in the 0.25- and 0.75-mg/kg/day groups in VBP15-003 dose escalated to either 2.0 or 6.0 mg/kg/day. The timing of dose escalations varied between participants and is shown in S1 Fig. Two participants in the 0.75-mg/kg/day VBP15-003 dose group escalated to 6.0 mg/kg/day, then later de-escalated to 2.0 mg/kg/day due to weight gain within the 12-month interim time period (S1 Fig; participants: 233409, 233102).

## Tolerability of dose escalation

Within the VBP15-LTE study, each participant could have his dose of vamorolone increased to a higher dose or decreased to a lower dose by the site investigator as necessitated clinically. Of the 11 participants in the 0.25-mg/kg/day dose group at entry in the VBP15-LTE, the vamorolone dose was increased to 2.0 mg/kg/day for 3 participants and to 6.0 mg/kg/day for 8 participants prior to the 12-month interim assessment (S1 Fig). The cumulative exposure to high-dose vamorolone (2.0 or 6.0 mg/kg/day) for those participants who were originally in the 0.25-mg/kg/day dose group ranged from 3 to 9 months (of the 18-month study period). Of the 12 participants in the 0.75-mg/kg/day dose group at entry in the VBP15-LTE, the vamorolone dose was increased to 2.0 mg/kg/day for 6 participants and to 6.0 mg/kg/day for 6 participants. The cumulative exposure to high-dose vamorolone for those participants who were originally in the 0.75-mg/kg/day dose group ranged from 9 to 11 months. Of the 12 participants in the 2.0-mg/kg/day dose group at entry in the VBP15-LTE, the dose remained at 2.0 mg/kg/day for 3 participants and was increased to 6.0 mg/kg/day for 9 participants. Two participants subsequently had their vamorolone dose decreased from 6.0 to 2.0 mg/kg/day due to weight gain. Of the 11 participants in the 6.0-mg/kg/day dose group at entry in the VBP15-LTE, all remained at this dose throughout the study period (S1 Fig).

## Efficacy evaluation of vamorolone-treated versus corticosteroid-naïve participants

Participants treated for 18 months with vamorolone (2.0 or 6.0 mg/kg/day) showed significant improvements in all measures of efficacy (Table 2). Paired *t* tests were significant for longitudinal improvements in all outcomes from baseline (TTSTAND velocity, $p = 0.012$ [95% CI 0.010, 0.068 event/second]; TTRW velocity, $p < 0.001$ [95% CI 0.220, 0.491 meters/second]; TTCLIMB velocity, $p = 0.005$ [95% CI 0.034, 0.105 event/second]; 6MWT, $p = 0.001$ [95% CI 31.14, 93.38 meters]; NSAA total score, $p < 0.001$ [95% CI 2.702, 6.662 points]). Group-matched corticosteroid-naïve participants from CINRG DNHS showed no change or slight improvements over this same time frame for TTRW velocity, TTCLIMB velocity, and TTSTAND velocity (6MWT and NSAA outcomes were not available in CINRG DNHS). ANCOVA comparisons between vamorolone-treated and corticosteroid-naïve participants did not show significant differences for TTSTAND (least squares [LS] mean 0.042 [95% CI −0.007, 0.091], $p = 0.088$), but showed significant differences favoring vamorolone for TTRW

**Table 2. Analyses of efficacy and safety outcome measures over 18 months, with comparison to corticosteroid-naïve DNHS participants.**

| Outcome and treatment group | n at baseline/18 months | Baseline value (SD) | 18-month value (SD)[1] | Change from baseline (SD) (95% 2-sided CI), paired t test p-value | LS mean difference (SE) (95% 2-sided CI), ANCOVA p-value |
|---|---|---|---|---|---|
| *Efficacy* | | | | | |
| **TTSTAND velocity (event/second)** | | | | | |
| Vamorolone | 23/22 | 0.206 (0.07) | 0.241 (0.076) | 0.039 (0.066) (0.010, 0.068) $p = 0.012$ | 0.042 (0.024) (−0.007, 0.091) $p = 0.088$ |
| Corticosteroid-naïve DNHS | 19/17 | 0.202 (0.055) | 0.205 (0.102) | −0.003 (0.083) (−0.046, 0.039) $p = 0.877$ | |
| **TTRW velocity (meters/second)** | | | | | |
| Vamorolone | 23/22 | 1.735 (0.331) | 2.061 (0.347) | 0.356 (0.306) (0.220, 0.491) $p < 0.001$ | 0.286 (0.09) (0.104, 0.469) $p = 0.003$ |
| Corticosteroid-naïve DNHS | 19/18 | 1.619 (0.483) | 1.717 (0.46) | 0.093 (0.281) (−0.047, 0.232) $p = 0.179$ | |
| **TTCLIMB velocity (event/second)** | | | | | |
| Vamorolone | 23/22 | 0.266 (0.134) | 0.331 (0.127) | 0.07 (0.08) (0.034, 0.105) $p = 0.001$ | 0.059 (0.026) (0.007, 0.111) $p = 0.027$ |
| Corticosteroid-naïve DNHS | 19/18 | 0.218 (0.098) | 0.242 (0.108) | 0.021 (0.089) (−0.023, 0.065) $p = 0.330$ | |
| **6MWT meters walked (meters)** | | | | | |
| Vamorolone | 20/19 | 343.2 (64.3) | 395.6 (69.7) | 62.2 (60.5) (31.14, 93.38) $p = 0.001$ | NA |
| **NSAA score (of 34)** | | | | | |
| Vamorolone | 23/22 | 19.9 (4.9) | 24.3 (4.7) | 4.7 (4.5) (2.702, 6.662) $p < 0.001$ | NA |
| *Safety* | | | | | |
| **Mean height percentile for age** | | | | | |
| Vamorolone | 23/22 | 29.19 (24.66) | 35.24 (29.82) | 6.92 (9.68) (2.622, 11.209) $p = 0.003$ | Versus vamorolone |
| Corticosteroid-naïve DNHS | 19/18 | 25.76 (21.37) | 27.16 (21.17) | 0.176 (11.72) (−5.653, 6.004) $p = 0.950$ | 6.72 (3.48) (−0.332, 13.78) $p = 0.061$ |
| Corticosteroid-treated DNHS | 68/68 | 20.09 (22.58) | 14.46 (22.69) | −5.63 (14.89) (−9.231, −2.026) $p = 0.003$ | 15.86 (3.70) (8.51, 23.22) $p < 0.001$) |
| Prednisone trial[1] | 12/12 | 29.89 (29.15) | 26.14 (24.21) | −3.76 (10.44) (−10.387, 2.877) $p = 0.238$ | 10.37 (3.86) (2.49, 18.25) $p = 0.012$ |
| **Mean BMI *z*-score** | | | | | |
| Vamorolone | 23/22 | 1.03 (0.56) | 1.46 (0.62) | 0.411 (0.615) (0.138, 0.683) $p = 0.005$ | Versus vamorolone |
| Corticosteroid-naïve DNHS | 19/18 | 0.70 (0.58) | 0.36 (0.77) | −0.345 (0.655) (−0.671, −0.019) $p = 0.039$ | 0.899 (0.204) (0.486, 1.31) $p < 0.001$ |
| Corticosteroid-treated DNHS | 68/67 | 0.98 (0.85)* | 1.13 (0.92) | 0.145 (0.518) (0.019, 0.272) $p = 0.025$ | 0.282 (0.146) (−0.01, 0.573) $p = 0.058$ |

*(Continued)*

**Table 2.** (Continued)

| Outcome and treatment group | n at baseline/18 months | Baseline value (SD) | 18-month value (SD)[1] | Change from baseline (SD) (95% 2-sided CI), paired t test p-value | LS mean difference (SE) (95% 2-sided CI), ANCOVA p-value |
|---|---|---|---|---|---|
| Prednisone trial[1] | 12/12 | 0.61 (1.27) | 1.068 (1.05) | 0.459 (0.407) (0.200, 0.718) p = 0.002 | 0.066 (0.193) (−0.328, 0.461) p = 0.733 |

[1]The 18-month value reflects outcome at 12 months of treatment, as this was the duration for the prednisone trial.

*Baseline indicates mean BMI at the beginning of the 18-month continuous treatment with corticosteroids. Participants may have been initiated on corticosteroids prior to this visit.

6MWT, 6-minute walk test; CI, confidence interval; DNHS, Duchenne Natural History Study; BMI, body mass index; LS, least squares; NA, not available; NSAA, North Star Ambulatory Assessment; SD, standard deviation; SE, standard error; TTCLIMB, time to climb 4 stairs; TTRW, time to run/walk 10 meters; TTSTAND, time to stand from supine.

velocity (LS mean 0.286 [95% CI 0.104, 0.469], $p = 0.003$) and TTCLIMB velocity (LS mean 0.059 [95% CI 0.007, 0.111], $p = 0.027$).

Results for measures in seconds are shown in S2 Table. Results from analysis of measures in seconds units showed significance for 18-month improvements in vamorolone-treated participants for TTRW ($p < 0.001$ [95% CI −1.53, −0.59 seconds]), but not TTSTAND ($p = 0.48$ [95% CI −1.90, 0.93 seconds]) or TTCLIMB ($p = 0.62$ [95% CI −2.67, 1.62 seconds]) due to severe outliers increasing variance. ANCOVA comparisons between vamorolone-treated and corticosteroid-naïve participants showed a significant difference favoring vamorolone for TTRW (LS mean −0.84 [95% CI −1.54, −0.14 seconds], $p = 0.02$), but not for TTSTAND (LS mean −1.15 [95% CI −2.87, 0.57 seconds], $p = 0.18$) or TTCLIMB (LS mean −0.34 [95% CI −3.28, 2.59 seconds], $p = 0.81$).

## Comparative efficacy of vamorolone dose groups

Participant-level data were analyzed graphically for the 4 vamorolone-treated groups relative to DNHS corticosteroid-naïve participants (Fig 1; left panels). Groups B, C, and D each showed improvements from baseline after 18 months of treatment in comparison to corticosteroid-naïve participants from CINRG DNHS, whereas group A outcomes were similar to those of corticosteroid-naïve participants. Of note, group A was treated for only 3 to 9 months with high-dose vamorolone (S1 Fig), and was also had a mean age 0.4 years older than that of the other groups at study entry (Group A, 5.2 ± 1.0 years; Groups B, C, and D, 4.8 ± 0.8 years). A cross-sectional comparison was carried out at 5.5–8.5 years of age (end of 18-month treatment period) (Fig 1; right panels), with visualization of the mean baseline of each of the 4 vamorolone groups and the DNHS corticosteroid-naïve ($n = 19$) and DNHS corticosteroid-treated comparators ($n = 68$). Vamorolone dose groups B, C, and D showed motor function outcomes that were similar to those of corticosteroid-treated DNHS participants. Corticosteroid-naïve participants showed poorer performance, as did vamorolone group A. These data suggest that the benefit of vamorolone at 2.0 or 6.0 mg/kg/day may be similar in magnitude to that of corticosteroid at 18 months of treatment.

## Pre-specified safety evaluation

Two measures of corticosteroid-associated safety concerns were pre-specified in the iSAP: growth deceleration (stunting of growth measured by change in mean height percentile for age) and body mass index (BMI) $z$-score. At baseline, the 3 groups (vamorolone, DNHS corticosteroid-treated, and DNHS corticosteroid-naïve) were generally short for age (mean 20th–

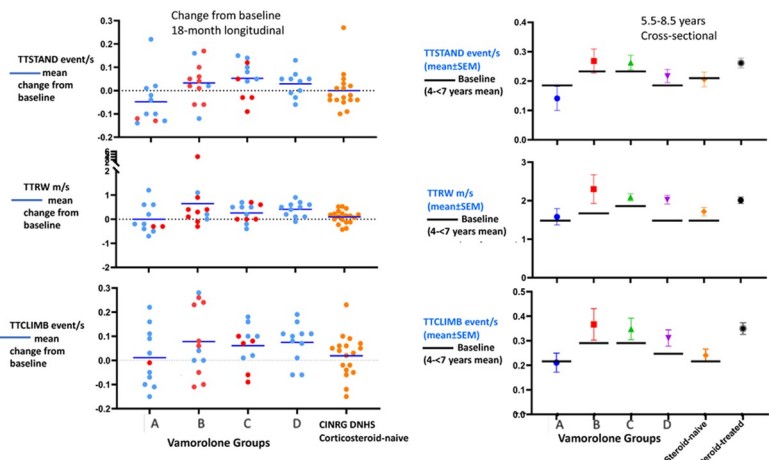

**Fig 1. Participant-level longitudinal data and aggregated cross-sectional data comparing vamorolone-associated efficacy to CINRG DNHS external comparators.** Left panels show participant-level change from baseline after an 18-month treatment period. Vamorolone group A was treated with 2.0 or 6.0 mg/kg/day for the last 3–9 months of the 18-month period, group B was treated with 2.0 or 6.0 mg/kg/day for the last 9–11 months, and groups C and D with 2.0 or 6.0 mg/kg/day for all 18 months. The specific dose of each participant at the end of the 18-month period is indicated (red = 2.0 mg/kg/day; blue = 6.0 mg/kg/day). Dose groups B, C, and D show mean improvements over baseline compared to matched corticosteroid-naïve participants from CINRG DNHS ($n$ = 19). Right panels show mean group cross-sectional analysis at age 5.5–8.5 years. The baseline mean is shown for each vamorolone-treated group (black line). The corticosteroid-treated natural history group ($n$ = 68) has no baseline shown, as the age at initiation of corticosteroids was variable. This panel shows improvement over baseline in vamorolone-treated groups B, C, and D, with the cross-sectional data suggesting an effect size similar to that of age-group-matched corticosteroid-treated participants in CINRG DNHS. CINRG, Cooperative International Neuromuscular Research Group; DNHS, Duchenne Natural History Study; SEM, standard error of the mean; TTCLIMB, time to climb 4 stairs; TTRW, time to run/walk 10 meters; TTSTAND, time to stand from supine.

29th height percentile for age). DNHS corticosteroid-naïve participants showed no change in growth trajectories over the 18-month period, whereas corticosteroid-treated participants showed the expected deceleration of growth seen with chronic treatment with corticosteroids (−5.63 mean change in height percentile) (Table 2). Vamorolone-treated participants showed a positive growth trajectory (+6.92 mean change in height percentile); this was not significantly different from the trajectory of the DNHS corticosteroid-naïve participants. Comparison of growth velocities of vamorolone-treated to DNHS corticosteroid-treated participants over the 18-month period showed a significant difference (LS mean 15.86 [95% CI 8.51, 23.22], $p <$ 0.001), and there was also a significant difference when comparing vamorolone 18-month treatment to prednisone trial 12-month treatment (LS mean 10.37 [95% CI 2.49, 18.25], $p =$ 0.012). This suggests that vamorolone treatment does not stunt growth, whereas corticosteroid-related growth stunting is a well-recognized safety concern.

For BMI $z$-score, the vamorolone-treated group had a normal mean BMI at baseline ($z$-score = 1.03), whereas DNHS corticosteroid-naïve and CINRG prednisone trial participants had a lower mean BMI at baseline ($z$-score = 0.70 and 0.61, respectively). CINRG corticosteroid-naïve participants showed a decrease in mean BMI over 18 months (change of $z$-score = −0.34). Participants in the CINRG prednisone clinical trial showed an increase of mean $z$-score of 0.46 over 12 months of treatment, and the vamorolone group showed increase of mean $z$-score of 0.41 over 18 months of treatment. CINRG DNHS participants who were treated with corticosteroids over an 18-month time period showed an increase of mean $z$-score of 0.15, but this group did not have measures prior to initiation of corticosteroids. The change in BMI was not significantly different between vamorolone- and corticosteroid-treated groups, whereas comparisons of drug-treated groups to corticosteroid-naïve participants

showed significant differences (Table 2). Stratification by original dose groups shows a general dose–response of increasing BMI (change from baseline to 18 months of treatment [kg/m$^2$]: group B, 0.5; group C, 1.11; group D, 2.55) with increasing vamorolone dose, although this was highly variable within all groups (S3 Table). This suggests that vamorolone may share the safety concern of weight gain with corticosteroids.

## Physician-reported AEs

Treatment-emergent AEs (TEAEs) have been published for the 2-week treatment MAD study (VBP15-002) [11] and the 24-week dose-finding extension study (VBP15-003) [14]. TEAEs were reported with similar incidence by participants in all 4 vamorolone groups. Several TEAEs commonly observed with chronic corticosteroid therapy were observed only in the 2.0-mg/kg/day group (abnormal behavior; 1 participant) and 6.0-mg/kg/day group (hypertrichosis [2 participants] and anxiety, abnormal blood cortisol level, Cushingoid habitus, and personality change [1 participant each]). The other reported TEAEs did not exhibit a dose-related incidence.

A Data and Safety Monitoring Board report on the VBP15-LTE study (3 December 2019; data cutoff 12 November 2019), covering all participants enrolled in the VBP15-LTE study (inclusive beyond the 12-month midpoint assessment) included 3 serious AEs (2 myoglobinuria events [in the same participant] and 1 pneumonia), all deemed unrelated to study drug. For all reported TEAEs, 402 were deemed unrelated to vamorolone, 37 were deemed remotely related, 29 were deemed possibly related, 11 were deemed probably related, and 3 were deemed definitely related. Of the 14 AEs probably and definitely related to vamorolone, 10 were weight gain, 2 were increased appetite, 1 was Cushingoid features, and 1 was irritability. A complete list of all TEAEs is provided for data cutoff beyond the 18-month last participant last visit, based on data cutoff of 12 November 2019 (S4 Table) (see S1 Fig for participant-level doses at time of TEAE).

We determined the incidences of physician-reported AEs typically associated with corticosteroid treatment for participants that had been treated with vamorolone 6.0 mg/kg/day (for any duration), taken from the March 2019 Data Safety Update Report (DSUR) (pharmacovigilance report). Incidence rates of Cushingoid features, weight gain, hirsutism, and behavior change were studied (Table 3). These rates were then compared to physician-reported AE incidences in the CINRG DNHS study [18], and a 12-month clinical trial of prednisone and deflazacort [20]. This comparison showed that there are lower rates of physician-reported Cushingoid features, weight gain, hirsutism, and behavior change in the vamorolone trial compared to published prednisone and deflazacort trials in boys with DMD.

**Table 3. Incidence of physician-reported adverse events.**

| Study | Treatment | n; mean age in years (SD)[1] | Cushingoid | Weight gain | Hypertrichosis/hirsutism | Behavior change |
|---|---|---|---|---|---|---|
| Vamorolone | 6.0 mg/kg/day vamorolone | n = 38; 4. 9 (0.9) | 2.6% | 13.2% | 0% | 0%[2] |
| Griggs et al. 2016 [20] | 0.9 mg/kg/day deflazacort | n = 68; 8. 8 (2.5) | 60.3% | 27.9% | 35% | 9% |
| | 0.75 mg/kg/day prednisone | n = 63; 8. 9 (2.9) | 77.8% | 34.9% | 44% | 14% |
| CINRG DNHS [18] | Deflazacort | n = 94 | 72% | 63% | NR | 33% |
| | Prednisone | n = 80 | 50% | 67% | NR | 30% |

Vamorolone data are from Data Safety Update Report 13 March 2019 (data cutoff 9 January 2019). Data shown are physician-reported adverse events.

[1]Mean age shown for vamorolone is a cross-sectional analysis; mean age shown for Griggs et al. is at baseline.

[2]No behavior change was reported, but 1 personality change, 1 sleep disorder, and 2 irritability were reported.

NR, Not reported.

## Discussion

A dose-ranging 24-week (6-month) study of vamorolone treatment in 4- to 7-year-old boys with DMD had shown dose-responsive improvements in motor function tests [14]. After completion of this study, participants were offered enrollment in a 2-year long-term extension study (VBP15-LTE) or transition to corticosteroid standard of care (deflazacort or prednisone). Here, we report interim findings in VBP15-LTE (18 months of vamorolone treatment). All participants (46 of 46) opted to enroll in the vamorolone long-term extension, suggesting high satisfaction with vamorolone treatment. Vamorolone-treated participants showed improvements from baseline in all 5 motor assessments over the 18-month treatment period (TTSTAND, TTRW, TTCLIMB, NSAA, and 6MWT) (Table 2). In contrast, group-matched steroid-naïve (non-treated) DMD participants in the CINRG DNHS study showed stable disease over a similar 18-month period. Comparisons of vamorolone-treated participants to CINRG DNHS non-treated participants showed that differences for TTSTAND were not significant, but significant vamorolone-related improvements were observed for TTRW velocity ($p = 0.003$) and TTCLIMB velocity ($p = 0.027$); data for NSAA and 6WMT were not available in the CINRG DNHS comparator group. Vamorolone has shown fewer morbidities than corticosteroids in mouse models of disease [3,5,21], but a comparative safety profile for vamorolone versus corticosteroids has not been previously reported in humans. Group-matched steroid-treated participants in the CINRG DNHS showed marked stunting of growth—a well-known safety concern with chronic deflazacort and prednisone treatment of children. In contrast, vamorolone-treated participants did not show any evidence of stunting of growth. Physicians reported fewer other corticosteroid-associated safety concerns in vamorolone-treated participants compared to published studies of deflazacort- and prednisone-treated DMD patients, including Cushingoid appearance, behavior change (mood disturbance), hirsutism, and weight gain.

While participating in the VBP15-LTE study, participants were permitted dose escalations and de-escalations at the discretion of families and their physicians. Most (74%; 34/46) opted to be treated with the highest dose permitted (6.0 mg/kg/day), and 26% with 2.0 mg/kg/day. There were 2 participants for whom the vamorolone dose was decreased from 6.0 to 2.0 mg/kg/day due to weight gain. DMD trial participants treated with 2.0 or 6.0 mg/kg/day vamorolone for the full 18-month period ($n = 23$) showed clinical improvement of all motor outcomes from baseline to month 18 (TTSTAND, $p = 0.012$; TTRW, $p < 0.001$; TTCLIMB, $p = 0.001$; 6MWT, $p = 0.001$; NSAA, $p < 0.001$). However, DMD patients in this young age range are, on average, stable or improving. Thus, it is important to compare improvements to non-treated participants.

The vamorolone clinical trials were conducted by the academic clinical trial network CINRG. The CINRG network had previously conducted a longitudinal natural history study of 551 DMD participants and healthy peers (CINRG DNHS) [16], with similar clinical evaluator methods and endpoints as utilized in the vamorolone trials. Pre-specified matching criteria were defined to provide group matching of corticosteroid-naïve and corticosteroid-treated cohorts selected from the CINRG DNHS to compare to vamorolone-treated participants over an 18-month time period. The comparator groups were similar to the vamorolone-treated groups at baseline, with slightly older ages in the CINRG DNHS study groups. These comparisons showed that DMD participants treated with vamorolone for 18 months (2.0 or 6.0 mg/kg/day) in comparison to corticosteroid-naïve participants did not show significant differences for TTSTAND velocity ($p = 0.088$), but did show significant improvement for TTRW velocity ($p = 0.003$) and TTCLIMB velocity ($p = 0.027$) (Table 2). Vamorolone treatment led to improvements in the 6MWT (mean +62.2 meters) and NSAA (mean +4.7 points), but these

outcomes were not measured over an 18-month interval in the CINRG DNHS, and, therefore, there was no group match comparator for these outcomes.

The cross-sectional graphical comparison of motor outcomes at the end of the 18-month treatment period (participants 5.5–8.5 years of age) shown in Fig 1 suggests that both the vamorolone-treated cohort (1 year or more treatment at 2.0 or 6.0 mg/kg/day: groups B, C, and D) and the CINRG DNHS corticosteroid-treated cohort had similar drug-related benefit relative to the CINRG DNHS corticosteroid-naïve cohort. Insufficient data were available to compare motor improvements with vamorolone to those natural history motor outcomes seen with specific corticosteroid regimens (e.g., daily prednisone and daily deflazacort). A multicenter placebo- and active-comparator (daily prednisone 0.75 mg/kg/day) controlled trial is currently being conducted (VBP15-004; NCT03439670).

Long-term treatment with corticosteroids (deflazacort and prednisone) is associated with a broad range of safety concerns that detract from patient quality of life [18,20]. In children, deceleration of linear growth is frequently seen with chronic corticosteroid treatment. Comparison of mean height percentile change over 18 months showed that corticosteroid treatment in CINRG DNHS participants led to stunting of growth (−5.63 percentile), whereas vamorolone treatment did not (+6.92 percentile) ($p < 0.001$) (Table 2). A double-blind clinical trial of prednisone versus deflazacort in DMD also found stunting of growth over a 12-month treatment period (−11.43 percentile for deflazacort; −7.04 percentile for prednisone) [20]. The lack of stunting of growth with high-dose vamorolone treatment is consistent with previously published bone turnover biomarker data, showing that corticosteroids lead to reductions in serum osteocalcin, but vamorolone does not [11,22,23]. These data suggest that vamorolone does not share stunting of growth with corticosteroids as a safety concern, and this may be a distinct advantage for children requiring chronic corticosteroid treatment. The physician-reported incidence of AEs was compared between the vamorolone trials, the corticosteroid-treated group in the CINRG DNHS [18], and the prednisone versus deflazacort trial [20] (Table 3). This comparison suggested a lower incidence of Cushingoid appearance, weight gain, hirsutism/hypertrichosis, and behavior change in vamorolone-treated DMD patients compared to corticosteroid-treated boys. Taken together, the data suggest that vamorolone treatment of DMD patients provides similar efficacy as corticosteroid treatment as assessed by motor function outcomes. Furthermore, this preliminary assessment indicates that vamorolone treatment resulted in a lower incidence of safety concerns typically associated with corticosteroid treatment. The BMI data from the 18-month extension in comparison to natural history data from the CINRG do not indicate that vamorolone-treated participants will be completely spared the side effects of weight gain, and 2 participants on 6.0 mg/kg/day had to de-escalate their dose to 2.0 mg/kg/day.

A limitation of these studies is their open-label study design and the indirect comparisons to previously published studies. However, long-term placebo arms in a rare pediatric disease such as DMD are considered unethical. Another limitation is that the CINRG DNHS corticosteroid-treated cohort had been on corticosteroids for a variable amount of time prior to the 18-month study interval. A double-blind study of 120 DMD participants enrolled into 4 groups is underway (vamorolone 2.0 and 6.0 mg/kg/day, placebo, and prednisone 0.75 mg/kg/day) (VBP15-004; NCT03439670), and this will provide Class I evidence for vamorolone efficacy and safety.

In conclusion, we present Class IV evidence that for boys with DMD, vamorolone treatment for 18 months shows possible efficacy compared to a natural history cohort of corticosteroid-naïve patients and appears to be well tolerated, with fewer safety concerns than typically seen with long-term standard-of-care corticosteroid treatment.

## Supporting information

**S1 TREND Checklist.**
(PDF)

**S1 Fig. Participant-specific dose levels in VBP15-LTE.** Each horizontal line indicates a participant in VBP15-LTE. *x*-Axis is date. The green star indicates the transition point of each participant from the time of VBP15-003 completion to VBP15-LTE enrollment. VBP15-003 was a 24-week dose-ranging study, with each group (*y*-axis) of 12 participants started at a specific dose (doses indicated by colors and legend; 0.25 mg/kg/day, 0.75 mg/kg/day, 2.0 mg/kg/day, 6.0 mg/kg/day). VBP15-LTE is a 2-year long-term extension study, but data presented here are from the midpoint (12 months of treatment in VBP15-LTE); the 12-month interim assessment time point is indicated by the red box in each participant. Dose escalations and dose de-escalations were permitted in VBP15-LTE at the discretion of the treating physician and the participant's family; dose changes are indicated for each participant by the change in color.
(DOCX)

**S1 iSAP. Interim statistical analysis plan for VBP15-LTE.**
(PDF)

**S1 Protocol. VBP15-LTE.** Amendment #1 for a 24-month Phase II open-label, multicenter long-term extension study to assess the long-term safety and efficacy of vamorolone in boys with Duchenne muscular dystrophy (DMD).
(PDF)

**S1 Raw Data.**
(CSV)

**S1 Table. Group-matching criteria for CINRG DNHS corticosteroid-treated and corticosteroid-naïve DMD participants.**
(DOCX)

**S2 Table. Summary of comparisons between vamorolone-treated (2.0 and 6.0 mg/kg/day) and corticosteroid-naïve CINRG DNHS participants (efficacy endpoints reported in seconds to achieve task).** CI, confidence interval; CINRG, Cooperative International Neuromuscular Research Group; DNHS, Duchenne Natural History Study; LS, least squares; SD, standard deviation; TTCLIMB, time to climb 4 stairs; TTRW, time to run/walk 10 meters; TTSTAND, time to stand from supine.
(DOCX)

**S3 Table. Mean (± SD) change in body mass index (BMI) for the VBP15-002/003 0.75-mg/kg/day (*n* = 12), 2.0-mg/kg/day (*n* = 11), and 6.0-mg/kg/day (*n* = 11) dose groups.**
(DOCX)

**S4 Table. Number of treatment-emergent adverse events by system organ class and preferred term.**
(DOCX)

## Author Contributions

**Conceptualization:** Eric P. Hoffman, Paula R. Clemens, Heather Gordish-Dressman, Benjamin D. Schwartz, Laurel J. Mengle-Gaw.

**Data curation:** Laurie S. Conklin, Paula R. Clemens, Lauren P. Morgenroth, Adrienne Arrieta, Maya Shimony, Phil Shale, Heather Gordish-Dressman, Laura Hagerty, Utkarsh J. Dang, Jesse M. Damsker.

**Formal analysis:** Laurie S. Conklin, Eric P. Hoffman, Mark Jaros, Phil Shale, Heather Gordish-Dressman, Laura Hagerty, Utkarsh J. Dang, Jesse M. Damsker, Benjamin D. Schwartz, Laurel J. Mengle-Gaw.

**Funding acquisition:** Eric P. Hoffman.

**Investigation:** Edward C. Smith, Jean K. Mah, Richard S. Finkel, Michela Guglieri, Mar Tulinius, Yoram Nevo, Monique M. Ryan, Richard Webster, Diana Castro, Nancy L. Kuntz, Craig M. McDonald.

**Methodology:** Eric P. Hoffman, Benjamin D. Schwartz.

**Project administration:** Paula R. Clemens, Jean K. Mah, Richard S. Finkel, Michela Guglieri, Mar Tulinius, Yoram Nevo, Monique M. Ryan, Richard Webster, Diana Castro, Nancy L. Kuntz, Laurie Kerchner, Lauren P. Morgenroth, Adrienne Arrieta, Maya Shimony, Heather Gordish-Dressman, Benjamin D. Schwartz, Laurel J. Mengle-Gaw.

**Supervision:** Laurie Kerchner, Heather Gordish-Dressman, Jesse M. Damsker, Benjamin D. Schwartz, Laurel J. Mengle-Gaw.

**Validation:** Mark Jaros.

**Writing – original draft:** Laurie S. Conklin, Eric P. Hoffman, Paula R. Clemens, Mar Tulinius, Heather Gordish-Dressman.

**Writing – review & editing:** Edward C. Smith, Laurie S. Conklin, Eric P. Hoffman, Paula R. Clemens, Jean K. Mah, Richard S. Finkel, Michela Guglieri, Mar Tulinius, Yoram Nevo, Monique M. Ryan, Richard Webster, Diana Castro, Nancy L. Kuntz, Laurie Kerchner, Lauren P. Morgenroth, Adrienne Arrieta, Maya Shimony, Mark Jaros, Phil Shale, Heather Gordish-Dressman, Laura Hagerty, Utkarsh J. Dang, Jesse M. Damsker, Benjamin D. Schwartz, Laurel J. Mengle-Gaw, Craig M. McDonald.

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
