## [Editor Report · Decision Letter 0]

17 Feb 2020

Dear Dr Smith, 

Thank you for submitting your manuscript entitled "Efficacy and safety of vamorolone in subjects with Duchenne muscular dystrophy: experience from an 18-month open-label study." for consideration by PLOS Medicine.

Your manuscript has now been evaluated by the PLOS Medicine editorial staff and I am writing to let you know that we would like to send your submission out for external peer review.

Kind regards,

Helen Howard, for Clare Stone PhD 

Acting Editor-in-Chief

PLOS Medicine 

plosmedicine.org

---

## [Decision Letter · Decision Letter 1]

20 Apr 2020

Dear Dr. Smith,

Thank you very much for submitting your manuscript "Efficacy and safety of vamorolone in subjects with Duchenne muscular dystrophy: experience from an 18-month open-label study." (PMEDICINE-D-20-00478R1) for consideration at PLOS Medicine. 

[LINK]

In light of these reviews, I am afraid that we will not be able to accept the manuscript for publication in the journal in its current form, but we would like to consider a revised version that addresses the reviewers' and editors' comments. Obviously we cannot make any decision about publication until we have seen the revised manuscript and your response, and we plan to seek re-review by one or more of the reviewers. 

We expect to receive your revised manuscript by May 11 2020 11:59PM. Please email us (plosmedicine@plos.org) if you have any questions or concerns.

We look forward to receiving your revised manuscript. 

Sincerely,

Emma Veitch, PhD

PLOS Medicine

On behalf of Clare Stone, PhD, Acting Chief Editor,

PLOS Medicine

plosmedicine.org

*Please revise your title according to PLOS Medicine's style. Your title must be nondeclarative and not a question. It should begin with main concept if possible. "Effect of" should be used only if causality can be inferred, i.e., for an RCT. Please place the study design ("A randomized controlled trial," "A retrospective study," "A modelling study," etc.) in the subtitle (ie, after a colon). Although the title and abstract currently state the study as an open-label study, this doesn't fully explain the study design (eg, whether randomized or not), and we'd suggest including that per the journal's usual style.

*Please structure your abstract using the PLOS Medicine headings (Background, Methods and Findings, Conclusions). The Methods and Findings sections should be one single sub-section, “Methods and findings”.

*At this stage, we ask that you include a short, non-technical Author Summary of your research to make findings accessible to a wide audience that includes both scientists and non-scientists. The Author Summary should immediately follow the Abstract in your revised manuscript. This text is subject to editorial change and should be distinct from the scientific abstract. Please see our author guidelines for more information: https://journals.plos.org/plosmedicine/s/revising-your-manuscript#loc-author-summary

*The referencing style used in the paper doesn't match PLOS Medicine's usual format (the paper uses author name-year system). Please change to numbered referencing in square brackets and then a numbered reference list. 

*One reviewer has said that part of the Results section reads like part of the Methods, however the editors would disagree with this point and would advise you to keep this information within the Results, however it could probably be presented more efficiently. 

*Although primarily designed to help support reporting of randomized trials, we'd still suggest that the authors consider using the CONSORT checklist to help improve reporting of the study. CONSORT checklist is available at: http://www.consort-statement.org/consort-statement/checklist

Comments from the reviewers:

Reviewer #1: See attachment

Michael Dewey

Reviewer #2: The abstract is well written.

The introduction is well written.

In the subjects and methods section, the obvious weakness of this trial, open label, was acknowledged.

The specific characteristics used to match the CINRG DNHS boys to the vamorolone trial boys should be spelled out in full, not relegated to the statement, "…and other characteristics (Supplemental Table 1)." Otherwise, there is no means to know the composition of the comparator group.

Results:

 - The first five paragraphs in the Results section read like a Methods section.

 - It remains difficult to read results presented in velocities (however statistically appropriate).

 - Opinion comments are best left to the Discussion section. There are numerous comments throughout the Results section.

Discussion:

This is written well.

It is key that you have mentioned the significant limitations associated with this trial design.

Figure 1: This is informative.

Table 1 is well laid out and comprehensively inclusive (mean, standard deviation, median, minimum, maximum). I am not sure all of this information is necessary in the table. Some could be placed in supplementary files.

Table 2 is well laid out and the numbers easy to see. However, the use of velocity measurements is not easily understood by most readers. Although statistically appropriate, it might be more reasonable to use raw data and discuss that velocity measurements confirmed (or reinforced) the results seen in the untransformed data.

Table 3 is well laid out and easy to understand.

New agents for the treatment of dystrophinopathies are important.

Corticosteroid-like agents, such as vamorolone, with the potential for fewer side effects are needed.

However, overall, this article is a very difficult read with multiple comparator groups (many of which are historical) and obtuse measurements across these multiple comparator groups.

Reviewer #3: The authors report on the 18-month safety and efficacy of an open-label trial of vamorolone in young boys with Duchenne muscular dystrophy with the aims of describing the safety and efficacy of this compound compared to external steroid naive and treated cohorts over a similar timeframe. Overall this study is useful to the field in providing an early view of signals of efficacy and safety of this compound as an alternative to traditional corticosteroid treatment options.

Major comments:

-There is inconsistent naming/labeling of groups/cohorts across text and figures, as well as differences in groups included in analysis that makes it challenging for the reader to easily follow and compare outcomes and safety across groups (specifically Supplemental Table 1). Additionally, the first paragraph states "Given the variable timing of dose escalations, we pre-specified that initial analyses of drug-related efficacy and safety would be limited to those subjects who had 18 months of treatment with 2.0 mg/kg/day vamorolone or more (dose group C + dose group D; n=23)." yet group A and B appear in Figure 1 and group B throughout the reported safety data.

-I would strongly recommend including steroid-treated functional data across tables and figures. Comparison of functional and safety data to a steroid-naive cohort has some utility, but the overall comparison to steroid-treated groups is more meaningful for clinicians in determining the best treatment options for their patients. While the variability in steroid treatment initiation is stated as the factor for not including this data, it would be preferential to include the data with the caveat that this variability in treatment duration may or may not impact findings. Additionally, it is included as a comparator for safety data and one would expect the same concern to stand for that comparison, yet it is included.

-Table 3 and the final paragraph of page 11 differs from the text in the previous paragraph on page 11 discussing AE's from the DSMB report (03Dec19). While this specific analysis is favorable for vamorolone, could be misleading to readers as weight gain appears to be quite common as evidenced by 10 AEs and similar increases in BMI. Additionally the authors comment on page 10 "This suggests that vamorolone may share the safety concern of weight gain with corticosteroids, although BMI does not distinguish between bone, muscle, and fat, yet adjusts for height. Each of these variables may be different for vamorolone-treatment and corticosteroid-treatment." The text "although BMI...." through the end of the next sentence should be removed as there is no evidence for this statement included in this study. Lastly, the vamorolone trial is open-label and being compared to randomized trials, so bias has the potential to significantly impact the difference in reporting.

Additional comments:

- Improvements in the 6MWT and NSAA are generally expected in this <7 age range. There are other external cohorts to compare these improvements which could be considered. A graph of individual patient improvements should be considered as it may be helpful in interpreting these changes. 

- The graphs on the right of Figure 1 appear to indicate more variability in performance in the vamorolone-treated cohorts than the steroid-treated group which makes the reader question whether there is a more variable response rate to vamorolone treatment than traditional corticosteroids? Could the authors comment on this variability as standard deviations in performance at baseline seem similar to the steroid-naive group.

[LINK]

---

## [Decision Letter · Decision Letter 2]

5 Jun 2020

Dear Dr. Smith,

Thank you very much for re-submitting your manuscript "Efficacy and safety of vamorolone in Duchenne muscular dystrophy: An 18-month non-randomized open-label extension study." (PMEDICINE-D-20-00478R2) for review by PLOS Medicine.

We appreciate your responses to the editorial and reviewer comments, and your responses to our questions concerning the trial analysis plan and data sharing. I have discussed the paper with my colleagues and it was also seen again by two reviewers. There are a few remaining editorial and production issues to be addressed in a revised version before we would be able to accept the paper for publication in the journal, and we do plan to seek the opinions of one or more of the reviewers.

[LINK]

We look forward to receiving the revised manuscript by Jun 12 2020 11:59PM. 

Sincerely,

Caitlin Moyer, Ph.D.

Associate Editor 

PLOS Medicine

plosmedicine.org

Requests from Editors:

1.Response to Editor Inquiry about interim analysis: [[Q1: Interim reporting: Please clarify whether this analysis was pre-planned, and if not, explain why this report describes only the first 12 months of a 24 month continuation study. On Page 116 (section 9.4) of your study protocol, it states that "No interim statistical analyses are planned." The response to Reviewer #1's question on interim reporting does not clarify this issue.

A1: As protocols are typically written and issued far ahead of statistical analysis plans (SAPs), it is not uncommon to have aspects of the SAPs at odds with original protocols. Such deviations are discussed by the study team (study chair, sponsor, medical monitor, clinical monitor, statisticians), and a judgement is made as to whether the changes are substantial enough to merit a Protocol Amendment (and re-review by all participating IRBs), or, alternatively, a listing of deviations of SAP vs. protocol is provided in each SAP under category "Summary of Statistical Analysis Changes to the Protocol". 

For the described VBP15-LTE open-label study, the study team discussed the option of an interim analysis, and agreed that it was important to address long-term safety data and efficacy data relative to external comparators. It was then discussed whether this deviation from Protocol warranted an Amendment to the VBP15-LTE protocol, or rather a listing in the "Summary of Statistical Analysis Changes to the Protocol" in the interim SAP. The study team agreed that given that this was an open-label study utilizing external comparators, and that interim analyses would not influence the integrity of the full 2-year study, that a note in the SAP "Summary of Statistical Analysis Changes to the Protocol" was the best approach. Attached is the fully executed interim SAP for VBP15-LTE, and the "Summary of Statistical Analysis Changes to the Protocol" is seen at 3.3 on Page 8 noting the deviation from the protocol.]]

Thank you for your response and for explaining this point. Please include this information in the text of the manuscript, or in the supporting information (called out in the main text of the manuscript) to clarify this.

2.Inclusion of pre-specified protocol: Thank you for clarifying the rationale behind the 12 month data analysis, and for sending us your interim SAP for reference. We note that it is marked “confidential” and request that you include a version (that still describes the planned and changed aspects of your analyses) as a supporting information file (e.g. S1 Interim SAP).

3.Data availability: We appreciate your willingness to make your data available. Please provide a link, or other access details, for data access, as part of the data availability statement. (Any means to provide data access is acceptable- The existing DYAD site you mentioned in your email message is fine provided it is updated with the data from this manuscript).

4. Abstract: Methods and Findings: Please present the TTSTAND results first in the order, as this is one of the primary study outcomes highlighted in the registered trial.

5.Abstract: Methods and Findings: Please include the population and setting of participants, and years during which the study took place.

6.Abstract: Methods and Findings: Please quantify the main results (with both 95% CIs and p values).

7.In the last sentence of the Abstract Methods and Findings section, please describe the main limitation(s) of the study's methodology.

8. Abstract: Conclusions: Please revise the sentence "Vamorolone ... improves gross motor outcomes...", to "We observed that vamorolone treatment was associated with improvements in some motor outcomes as compared with corticosteroid-naïve individuals". To clarify that not all motor outcomes measured were found to be significantly improved.

9. Abstract: Conclusions: Please clarify the statements in the conclusion regarding adverse effects- something similar to the following may be helpful: “We found that fewer physician-reported adverse events occurred with vamorolone than have been reported for treatment with prednisones and deflazacort.” 

10.Author Summary: Please structure the Author Summary using bullet points rather than paragraph format. Please see our author guidelines for more information: https://journals.plos.org/plosmedicine/s/revising-your-manuscript#loc-author-summary

In the author summary, the section “Why Was This Study Done?” should provide the rationale for investigating both the motor function outcomes and safety concerns (as you report results relevant to each).

In the author summary section “What did the authors do and find?” please make it clear that not all measured components of motor function (e.g. TTSTAND) demonstrated improvement (as in when compared with the external comparison group)

In the author summary, please clarify what is meant by “decreased safety concerns” (do you mean fewer reported AEs, for example?)

11. Results: When presenting the results, please explicitly present the TTSTAND findings first, as this is one of the primary outcomes proposed in the trial registry. The other motor outcomes are registered as secondary outcomes, and the results should be presented and discussed in that order for clarity.

12.Discussion: Please present and organize the Discussion as follows: a short, clear summary of the article's findings; what the study adds to existing research and where and why the results may differ from previous research; strengths and limitations of the study; implications and next steps for research, clinical practice, and/or public policy; one-paragraph conclusion.

13. Discussion: Please revise the sentence in the Discussion stating “These comparisons showed that DMD participants treated with vamorolone for 18 months (2.0 or 6.0 mg/kg/day) showed improvements in all motor outcomes studied, in comparison to corticosteroid-naïve participants for TTRW (p=0.003) and TTCLIMB velocity (p=0.027) (Table 2).” Although you mention the TTSTAND findings in the following sentence, this sentence is misleading as you mention improvements in all motor outcomes studied. As TTSTAND is your primary registered motor outcome, please present your discussion of that outcome first.

14.Checklist: Thank you for including the TREND checklist. When completing the checklist, please use section and paragraph numbers to refer to locations in the text, rather than page numbers, as page numbers are subject to change. Please add the following statement, or similar, to the Methods: "This study is reported as per the Transparent Reporting of Evaluations with Nonrandomized Designs (TREND) guideline (S1 Checklist)."

15. Figure 1: Please present TTSTAND results first (top) and please provide the abbreviations for TTSTAND, TTCLIMB and TTRW in the figure legend.

16. Table 1: In the legend, please define abbreviations for CINRG, DNHS.

17. Table 2: Please include units for velocity measures. Please report the TTSTAND results first in the table, as this was your primary outcome. Please define abbreviations for DNHS, LS, NSAA, 6MWT, TTSTAND, TTCLIMB, TTRW, BMI, SD, SE, CI, and any other abbreviations used.

18. Table 3: Please define the abbreviation “Nr”

19. Supporting information Figure 1: Please provide a descriptive legend for the figure, including the meaning behind the green stars, the red squares, and the axis labels.

20. S2 Table: Please provide (in a descriptive legend for the table) a brief description of the units of the measures for each test (seconds to what endpoint? Seconds to complete task/achieve criteria, etc). Please also define abbreviations for TTSTAND, TTCLIMB, TTRW, and present the TTSTAND outcomes first.

Comments from Reviewers:

Reviewer #1: The authors have addressed most of my points but there still remain a couple of issues.

I suggested a sensitivity analysis of the time data alongside the use of velocities. I am afraid I do not completely understand the authors response in the rebuttal letter. However this was only a suggested sensitivity analysis so it is perhaps not necessary to push the issue.

More seriously the authors do not seem to have understood my point about the premature end of the study. My apologies if I was not sufficiently clear here but if a study planned to last for 24 months is stopped and reported prematurely then this raises the issue of how that decision was made. Was this made independently of the investigators by the oversight committee? Was an interim analysis planned and how was that carried out? The desire for transparency about this is driven by the fact that in the Bad Old Days people had preliminary peeks at the data and so used up all their alpha. I am not suggesting the authors did this but for full transparency we need to know what happened.

Michael Dewey

Reviewer #3: Thank you for your thorough response to all comments and included edits. No additional concerns with this revision.

[LINK]

---

## [Editor Report · Decision Letter 3]

25 Aug 2020

Dear Dr. Smith, 

On behalf of my colleagues and the academic editor, Dr. Lindsay Alfano, I am delighted to inform you that your manuscript entitled "Efficacy and safety of vamorolone in Duchenne muscular dystrophy: An 18-month interim analysis of a non-randomized open-label extension study." (PMEDICINE-D-20-00478R3) has been accepted for publication in PLOS Medicine. 

PRODUCTION PROCESS

PRESS

PROFILE INFORMATION

Thank you again for submitting the manuscript to PLOS Medicine. We look forward to publishing it. 

Best wishes, 

Caitlin Moyer, Ph.D.

Associate Editor 

PLOS Medicine

plosmedicine.org